# Effects of Bempedoic Acid in Acute Myocardial Infarction in Rats: No Cardioprotection and No Hidden Cardiotoxicity

**DOI:** 10.3390/ijms24021585

**Published:** 2023-01-13

**Authors:** Tamás G. Gergely, Gábor B. Brenner, Regina N. Nagy, Nabil V. Sayour, András Makkos, Csenger Kovácsházi, Huimin Tian, Rainer Schulz, Zoltán Giricz, Anikó Görbe, Péter Ferdinandy

**Affiliations:** 1MTA-SE System Pharmacology Research Group, Department of Pharmacology and Pharmacotherapy, Semmelweis University, H-1089 Budapest, Hungary; 2Institute of Physiology, Justus Liebig University Giessen, 35390 Giessen, Germany; 3Pharmahungary Group, H-6722 Szeged, Hungary

**Keywords:** bempedoic acid, hidden cardiotoxicity, cardioprotection, acute myocardial infarction, ischemia/reperfusion injury, hyperlipidemia, nexletol, nilemdo, arrhythmias

## Abstract

Lipid-lowering drugs have been shown to have cardioprotective effects but may have hidden cardiotoxic properties. Therefore, here we aimed to investigate if chronic treatment with the novel lipid-lowering drug bempedoic acid (BA) exerts hidden cardiotoxic and/or cardioprotective effects in a rat model of acute myocardial infarction (AMI). Wistar rats were orally treated with BA or its vehicle for 28 days, anesthetized and randomized to three different groups (vehicle + ischemia/reperfusion (I/R), BA + I/R, and positive control vehicle + ischemic preconditioning (IPC)) and subjected to cardiac 30 min ischemia and 120 min reperfusion. IPC was performed by 3 × 5 min I/R cycles before ischemia. Myocardial function, area at risk, infarct size and arrhythmias were analyzed. Chronic BA pretreatment did not influence cardiac function or infarct size as compared to the vehicle group, while the positive control IPC significantly reduced the infarct size. The incidence of reperfusion-induced arrhythmias was significantly reduced by BA and IPC. This is the first demonstration that BA treatment does not show cardioprotective effect although moderately reduces the incidence of reperfusion-induced arrhythmias. Furthermore, BA does not show hidden cardiotoxic effect in rats with AMI, showing its safety in the ischemic/reperfused heart.

## 1. Introduction

Unexpected adverse effects are the leading cause of drug withdrawals from the market. Adverse cardiac effects account for 20% of all adverse toxic events, which makes them the main reason for the failure of clinical trials during drug development [1]. Currently, in preclinical cardiac safety testing, the effects of drug candidates on the cardiovascular system are only tested in healthy animals and tissues, while the presence of comorbidities and pharmacological treatment of these diseases are neglected. ‘Hidden cardiotoxicity’ is a phenomenon where the cardiac toxicity caused by a drug can be observed only in a diseased condition, such as ischemia/reperfusion injury and/or in the presence of cardiovascular-disease-related comorbidities [2,3].

Hypercholesterolemia is a common comorbidity in cardiovascular diseases, and an elevated level of low-density lipoprotein cholesterol (LDL-C) is a major risk factor for the development of atherosclerotic cardiovascular disease (ASCVD) [4]. Due to this, clinical guidelines recommend intensive lowering of LDL-C in high-risk patients [5], and the 3-hydroxy-3-methyl-glutaryl coenzyme A (HMG-CoA) reductase inhibitor statins are the first-line therapy in the treatment of dyslipidemias. Although there is a clear cardiovascular benefit of statin use, non-adherence to statin therapy is quite common [6], in part due to misinformation on lipid-lowering therapy [7]. Statin intolerance occurs in 9.1% of patients [8], mainly in the form of skeletal-muscle-related adverse events [9]. Moreover, in our previous study, lovastatin showed hidden cardiotoxic effect in rats, manifested as blocking the cardioprotective effect of ischemic conditioning [2,10]. Early discontinuation of the therapy increases risk of myocardial infarction (MI) and death from CVD [11]; thus, the development of novel, non-statin lipid-lowering agents is of high clinical relevance.

Bempedoic acid (BA) is a novel, first-in-class lipid-lowering small molecule, inhibiting ATP-citrate lyase (ACL), which results in the attenuation of de novo cholesterol synthesis. Bempedoic acid was approved by the FDA in 2020 after clinical trials showing efficacy in reducing LDL-C levels. It has been shown that bempedoic acid is a prodrug activated in the liver, but not in the skeletal muscle [12]; thus, it potentially leads to fewer muscle-related adverse effects compared to statins. BA is indicated currently for the reduction of LDL-C in patients with statin intolerance, or as an additive treatment with statins.

Statins and other lipid-lowering therapies, such as PCSK9 inhibitors, have been shown to exert pleiotropic effects in the cardiovascular system beyond the reduction in cholesterol, including a role in myocardial ischemia/reperfusion injury [13,14]. However, as cardiac safety of novel drug candidates is not tested routinely in animals with cardiac pathologies, little is known about the effects of bempedoic acid in the ischemic heart. Therefore, in this study, we aimed to investigate the potential hidden cardiotoxic or cardioprotective effects of bempedoic acid in a rat model of cardiac ischemia/reperfusion injury in vivo.

## 2. Results

### 2.1. Chronic Bempedoic Acid Treatment Does Not Affect Cardiac Function or Ventricular Diameters

Male Wistar rats were treated with bempedoic acid (BA) or vehicle for 28 days orally (Figure 1). After 28 days of treatment, animals underwent echocardiography, to investigate the effect of chronic bempedoic acid treatment on cardiac functional and morphological parameters. Bempedoic acid did not affect systolic function, as shown by normal ejection fraction and fractional shortening (Figure 2). Furthermore, diastolic function was also preserved, as shown by similar E/e′ ratios. Left ventricular morphology and hypertrophy were assessed by measurement of ventricular diameters and left ventricular mass during echocardiography. Bempedoic acid did not alter the left ventricular diameters nor the total mass of the ventricle. Complete results from the echocardiographic measurements are shown in Appendix A. As a summary, these data show that the use of bempedoic acid does not affect cardiac function or morphology in healthy animals.

### 2.2. Bempedoic Acid Pretreatment Does Not Influence Ischemia/Reperfusion-Induced Mortality

In order to investigate the potential hidden cardiotoxic and/or cardioprotective effects of bempedoic acid, we have performed occlusion of the left anterior descending (LAD) coronary for 30 min, followed by 120 min of reperfusion to induce cardiac ischemia/reperfusion injury (Figure 1). As a positive control for cardioprotection, ischemic preconditioning (IPC) was elucidated in a group of vehicle-treated animals by 3 × 5 min of occlusion and reperfusion of the LAD. We investigated I/R-induced mortality between groups, as a measure of hidden cardiotoxic and/or cardioprotective effects.

No statistically significant difference was found between groups in mortality during I/R surgery (I/R + vehicle: 19.23%, I/R + BA: 29.17%, IPC + BA: 10%, Chi-square test: n.s., *n* = 26–30/group). In the I/R + vehicle group, out the 26 performed surgeries 0 had to be excluded, 21 animals survived the surgery and five died. In the I/R + BA group, out of the 26 performed surgeries two had to be excluded due to predefined exclusion criteria, 17 animals survived and seven animals died. In the IPC + vehicle group, out of the 35 performed surgeries seven had to be excluded, 27 survived and three animals died. Altogether, this suggests that bempedoic acid pretreatment does not exacerbate nor protect from I/R-induced acute mortality in this model.

### 2.3. Bempedoic Acid Pretreatment Does Not Affect Infarct Size after Ischemia/Reperfusion Injury

For endpoints of I/R injury, first we measured myocardial infarct size, expressed as a proportion of the total LV area exposed to ischemia (area at risk, AAR). The AARs did not differ between groups (I/R + vehicle: 29.61% ± 3.08, I/R + BA: 33.90% ± 1.95, IPC + vehicle: 27.29% ± 2.30, Kruskal–Wallis test: n.s., *n* = 11–15/group).

Chronic pretreatment with BA did not influence infarct size compared to the vehicle group, showing no hidden cardiotoxic or cardioprotective effects, while the positive control IPC significantly reduced it (Figure 3).

### 2.4. Bempedoic Acid Pretreatment Reduces Reperfusion-Induced Arrhythmias

Arrhythmia analysis was performed by evaluating the ECG recordings during ischemia and the first 15 min of reperfusion, according to the Lambeth Conventions [15]. The most severe arrhythmia was visualized using the arrhythmia map in 5 min periods (Figure 4). The severity of arrhythmias during ischemia and reperfusion was scored according to the scoring system described by Curtis et al. [16]. The arrhythmia score during ischemia was not affected by bempedoic acid compared to vehicle treated group, while IPC significantly decreased it. However, during reperfusion, the arrhythmia score was significantly reduced both by bempedoic acid treatment and IPC, compared to the I/R + vehicle group (Figure 5). Incidence and duration of arrhythmias are shown in Appendix A. The decrease in arrhythmia score during reperfusion by bempedoic acid can be attributed to the reduction in the incidence of non-sustained ventricular tachycardias (NSVTs).

## 3. Discussion

In this study, we investigated the potential cardioprotective and/or hidden cardiotoxic effect of bempedoic acid in a rat model of acute myocardial infarction. Here we show that BA does not affect infarct size after I/R injury in rats; however, it moderately decreases reperfusion-induced arrhythmias. This is the first demonstration that chronic administration of bempedoic acid does not show cardioprotective or hidden cardiotoxic effects in rats with acute myocardial infarction.

Bempedoic acid is a novel, first-in-class inhibitor of ATP-citrate lyase (ACL), approved for the treatment of hypercholesterolemia. In the CLEAR (Cholesterol Lowering via BEmpedoic Acid, an ACL-inhibiting Regimen) program, the safety and efficacy of bempedoic acid have been demonstrated in randomized, placebo-controlled trials, including in statin-intolerant patients or as an adjunct to maximally tolerated statin therapy [17,18,19,20]. However, the long-term effects of bempedoic acid on cardiovascular outcomes are currently under investigation in the ongoing CLEAR Outcomes study, which includes a total of 14,014 patients with statin intolerance and established or high risk of cardiovascular disease [21].

As current preclinical guidelines only require testing new compounds on young, healthy animals during drug development, hidden cardiotoxic effects of drugs are not possible to detect in the early phase of development [2]. The mechanisms of hidden cardiotoxicity may include the drug-induced enhancement of cell death or pro-arrhythmic processes during myocardial I/R injury, as well as the inhibition of cardioprotective signaling pathways. Several drugs have been shown to interfere with myocardial I/R and the cardioprotective signaling pathways, thus showing potential hidden cardiotoxic effects, including nitrates (with the development of nitrate tolerance), adenosine triphosphate-dependent potassium inhibitor antidiabetic drugs, cyclooxygenase-2 (COX-2) inhibitors and statins [13]. Investigation of the cardiac effects of novel compounds in the diseased myocardium, e.g., after I/R injury, may uncover cardiac adverse effects of drugs that otherwise remain hidden for years after market authorization. This is best illustrated by the case of the COX-2 inhibitor rofecoxib, which was withdrawn from the market due to unexpected cardiovascular adverse effects, only observed in post-marketing studies [22]. Previously, our group has shown that using preclinical models of I/R injury could be suitable to detect hidden cardiotoxic effects of drugs. In a rat model of myocardial I/R injury, 28 days of rofecoxib pretreatment was found to increase the mortality and irreversible ventricular fibrillations during cardiac ischemia [3]; thus, it showed hidden cardiotoxic effects according to our definition, while we have found that the PPAR-γ agonist antidiabetic drug, rosiglitazone, does not show major hidden cardiotoxic effects, but it interferes with the antiarrhythmic effect of ischemic preconditioning [23].

Importantly, Kocsis et al. have found that chronic administration of the lipid-lowering drug, lovastatin, abolishes the cardioprotective effect of ischemic conditioning, therefore exhibiting hidden cardiotoxic effects [10]. In our current study, bempedoic acid did not show hidden cardiotoxic properties, as chronic pretreatment with bempedoic acid did not influence infarct size after acute I/R injury and did not exacerbate I/R-induced arrhythmias or mortality. The lack of interference with infarct size suggests chronic bempedoic acid treatment is safe to use in patients with high risk for acute myocardial infarction. BA treatment did not affect systolic or diastolic myocardial function, further showing the good safety profile of BA. The absence of proarrhythmic effect is in line with clinical findings showing that bempedoic acid does not affect QT length and cardiac repolarization in healthy human volunteers [24]. However, no data is available on the electrophysiological effects of bempedoic acid in the diseased myocardium, where pathological states, e.g., ischemia and reperfusion, can interact with the effects of drugs, revealing effects that are not evident in patients with no myocardial disease [2,3]. Further studies, using preclinical arrhythmia models, such as ex vivo simulated I/R injury [3], animals with reduced repolarization reserve [25,26], or electrocardiogram-based scoring systems [27], may be required to further demonstrate the cardiac safety of this novel antihyperlipidemic drug.

On the other hand, lipid-lowering drugs have been shown to exert beneficial, pleiotropic effects on the cardiovascular system beyond the decrease in cholesterol, further warranting the investigation of this new lipid-lowering drug in models of cardiac disease. Statins have been shown to decrease infarct size in different models of cardiac I/R, including pigs [28], rabbits [29] and rats [10,30]. Furthermore, PCSK9 inhibitors have also been shown to reduce infarct size in I/R models in rats [31,32], although in one study the cardioprotective effect was only evident ex vivo but not in PCSK9^-/-^ mice in vivo [33]. In our current study, cardioprotective effect regarding acute infarct size reduction was not found with chronic bempedoic acid treatment. In clinical studies, reduction of hsCRP suggests a potential anti-inflammatory component to the effect of bempedoic acid [34], similarly to statins. In this regard, investigation of bempedoic acid in chronic myocardial infarction models may be still relevant, as inflammatory processes play a major role in post-MI heart failure development [35]. Interestingly, we have found that bempedoic acid decreased the occurrence of early reperfusion-induced arrhythmias, mainly non-sustained ventricular tachycardias (NSVTs). This suggests that there is a potential protective effect of bempedoic acid in I/R injury; however, further studies are needed to confirm this effect and investigate the underlying mechanisms. This alleviating effect of bempedoic acid regarding I/R-induced arrhythmias is similar to other lipid-lowering drugs in preclinical studies. Statins have been shown to exert antiarrhythmic effects [36], including decrease of arrhythmias during I/R injury in rats [37,38]. Furthermore, in a preclinical study, pretreatment of rats undergoing I/R injury with a PCSK9 inhibitor decreased the arrhythmia score as well [31].

In summary, here we demonstrate that chronic bempedoic acid treatment does not show cardioprotective effect in a rat model of acute I/R injury, although it moderately decreases reperfusion-induced arrhythmias. Furthermore, bempedoic acid does not show hidden cardiotoxic effects, demonstrating its safety in the ischemic/reperfused heart.

## 4. Materials and Methods

### 4.1. Animals and Materials

All procedures were approved by the National Scientific Ethical Committee on Animal Experimentation and the Semmelweis University’s Institutional Animal Care and Use Committee (H-1089 Budapest, Hungary) in accordance with NIH guidelines (National Research Council (2011), Guide for the Care and Use of Laboratory Animals: Eighth Edition) and permitted by the government of Food Chain Safety and Animal Health Directorate of the Government Office for Pest County (project identification code: PE/EA/1912-7/2017; date of approval: November 2017).

Male Wistar rats (ranging from 92 g to 150 g in the beginning of the treatment) were used in the study, obtained from Toxi-Coop Zrt. (Budapest, Hungary). The animals had an acclimatization period of at least 1 week prior to experiments. The animals were housed in a temperature- (22 ± 2 °C), and humidity-controlled room at a 12 h light/dark cycle in individually ventilated cages with shelters, holding 2 animals per cage, and had free access to laboratory chow and drinking water ad libitum. Animals were not fasted before surgery.

Bempedoic acid was purchased from Cayman Chemicals (Ann Arbor, MI, USA). Evans blue dye (#E2129) and triphenyltetrazolium chloride (#T8877) were purchased from Sigma (St. Louis, MO, USA).

### 4.2. Study Design

The schematic protocol for the study is shown in Figure 1. The animals were treated with bempedoic acid (30 mg/kg, *n* = 26) or its vehicle, 1% hydroxyethylcellulose (*n* = 61), once daily by oral gavage for 28 days. The dose of bempedoic acid was chosen based on previous studies, which showed that four weeks of treatment in rats resulted in plasma levels similar to clinical findings, while no drug-related toxicities were found [39]. None of the animals died or were excluded during the treatment period. Animals were weighed daily during this period; body weights are shown in Appendix A. After 28 days of treatment, animals underwent echocardiography to determine the cardiac function of animals after chronic bempedoic acid treatment. Following the echocardiography, surgical induction of cardiac ischemia and reperfusion (I/R) was performed by occlusion of the left anterior descending coronary artery (LAD) for 30 min, followed by 120 min of reperfusion, while in one group ischemic preconditioning (IPC) was performed by three cycles of 5 min I/R prior to the index ischemia. Animals were randomized into the following surgical groups: vehicle + I/R (*n* = 26), bempedoic acid + I/R (*n* = 26) and vehicle + IPC (*n* = 35) as positive control. Animals were randomized into groups in a way that on each surgical day at least one animal from all groups were operated. The order of animals was randomized for each surgical day separately. For investigation of cardiac function via echocardiography, the minimum number of animals needed to be included was determined by a priori power calculation. Echocardiographic measurements were performed at the beginning of each surgical day until reaching the sufficient number of animals for both groups; however, due to the uneven distribution of BA and vehicle treated animals, the final number of measurements is different between the two groups (Figure 2).

### 4.3. Echocardiography

Rats were anesthetized with 60 mg/kg pentobarbital intraperitoneally and placed on heating pads maintaining 37 °C body temperature, with continuous monitoring via rectal probe. Echocardiographic analysis was performed with Vevo 3100 high-resolution in vivo imaging system (Fujifilm VisualSonics, Toronto, ON, Canada) using an ultrahigh frequency MX250 transducer (30 MHz, 55 frames per second), by an operator blinded to the study groups. On two-dimensional recordings of the short-axis at the mid-papillary muscle level, measured parameters included left ventricular internal diameter in systole and diastole (LVIDs and LVIDd, respectively) and LV anterior and posterior wall thickness in diastole (LVAWd and LVPWd, respectively). End-diastolic and end-systolic LV areas were measured from short- and long-axis two-dimensional B-mode recordings. Diastolic parameters were measured in the apical four-chamber view. Pulse-wave Doppler and tissue Doppler were used to determine early mitral inflow velocity (E) and mitral annular early diastolic velocity (e’), respectively. Fractional shortening (FS) was calculated as [(LVIDd − LVIDs)/LVIDd] × 100. End-diastolic (LVEDV) and end-systolic (LVESV) LV volumes were calculated from the rotational volumes of the left ventricular trace at the diastole and systole around the long axis line of the spline. Stroke volume (SV) was calculated as LVEDV − LVESV. Ejection fraction (EF) was determined as (SV/LVEDV) × 100. Cardiac output was calculated as SV × HR/1000). LV mass was calculated according to a cubic formula, suggested by Devereux et al. [40] and modified for rodents [41]: LV mass = 1.04 [(LVIDd + LVAWd + LVPWd)^3^- LVIDd^3^] × 0.8 + 0.6). Echocardiographic recordings were evaluated via VevoLAB software (Fujifilm VisualSonics, Toronto, ON, Canada) by an evaluator blinded to the study groups.

### 4.4. Surgical Induction of Acute Ischemia/Reperfusion Injury

Rats were anesthetized with 60 mg/kg pentobarbital intraperitoneally (Produlab Pharma, Raamsdonksweer, The Netherlands). The absence of pedal reflex was considered as being deep surgical anesthesia. Anesthesia was maintained by supplying half dose pentobarbital i.p. as required when plantar reflex could be elicited through regular paw pinch monitoring. Body surface electrocardiogram (ECG) was monitored throughout the experiments by using standard limb leads (AD Instruments, Bella Vista, Australia). The cannulated right carotid artery was used for the measurement of mean arterial blood pressure (MAP, AD Instruments, Bella Vista, Australia), and fluid supplementation occurred with saline containing 10 IU kg^−1^ heparin. The core body temperature was maintained at physiological temperature with a heating pad (Harvard Apparatus, Holliston, MA, USA). After orotracheal intubation, rats were ventilated with a rodent ventilator (Ugo-Basile, Gemonio, Italy) with room air in a volume of 6.2 mL/kg and frequency of 69 ± 3 breath/min.

Myocardial ischemia was induced by the occlusion of the left anterior descending coronary artery (LAD). A 5-0 Prolene suture (Ethicon, Johnson & Johnson, Budapest, Hungary) was looped around the LAD, by an operator blinded to study groups. Reversible myocardial ischemia was induced by tightening a snare around the LAD. After occlusion of the LAD, the presence of myocardial ischemia was confirmed by the appearance of ST-segment changes, I/R-induced arrhythmias and visible pallor of the myocardial regions distal to the occlusion. After 30 min of LAD occlusion, 120 min of reperfusion was induced by relieving the snare. Reperfusion was confirmed by ST-segment normalization, occurrence of early reperfusion arrhythmias and conspicuous hyperemia of the reperfused cardiac region. To prevent coagulation, the animals received intraperitoneal injections of 100 IU kg^−1^ heparin at 35th, 65th and 185th minutes of experiments.

Predefined exclusion criteria during surgery were the following: iatrogenic or technical error leading to the death of the animal, severe bleeding during the surgery, lack of confirmed ischemia after LAD occlusion by visual inspection or ECG alterations, lack of confirmed reperfusion after the release of the occlude by normalization of ECG alterations or visual inspection.

### 4.5. Infarct Size Measurement

After 120 min of reperfusion, hearts were excised and perfused for 2 min with oxygenated Krebs–Henseleit solution (in mM: NaCl 118, KCl 4.7, MgSO_4_ 1.2, CaCl_2_ 1.25, KH_2_PO_4_ 1.2, NaHCO_3_ 25, and glucose 11) at 37 °C in Langendorff mode to remove blood from the tissue, LAD was re-occluded, and the area at risk (AAR) was negatively stained with Evans blue dye through the ascending aorta. For the assessment of viable myocardial tissue, 2 mm-thick slices were cut and incubated in 1% triphenyltetrazolium chloride at 37 °C for 14 min. The slices were weighed and scanned. Planimetric analyses were performed by two independent, blinded investigators with InfarctSize 2.4b software (Pharmahungary Group, Budapest, Hungary). Area at risk (AAR) was expressed as the proportion of the left ventricular area, and the infarct size as the proportion of the AAR, and then areas were normalized to the mass of each slice.

### 4.6. Arrhythmia Analysis

The severity and duration of I/R-induced arrhythmias were analyzed by independent investigators in a blinded fashion. Continuous ECG records of each animal were evaluated according to the Lambeth conventions and quantified by using the ‘score A’ as previously described by Curtis et al. [15,16]. The whole 30 min period of ischemia and the first 15 min of reperfusion were quantified separately. Visual representation of the most severe arrhythmia during the 30 min of ischemia and first 15 min of reperfusion is shown by the arrhythmia map, where the 5 min periods were colored according to the most severe arrhythmia type.

### 4.7. Mean Arterial Pressure and Heart Rate Measurement

Blood pressures and heart rates were continuously recorded during the surgery. Appendix A shows the average blood pressures before surgery (baseline), during ischemia and reperfusion.

### 4.8. Statistical Analysis

The statistical analysis was performed with GraphPad Prism software (version 8.0.1). *p* < 0.05 was considered significant in all analysis. Normal distribution of data was tested by Shapiro–Wilk normality test. For comparisons between two groups, either a parametric two-tailed Student’s *t*-test or a nonparametric Mann–Whitney U-test was performed. One-way ANOVA followed by Tukey’s post hoc test or Kruskal–Wallis test followed by Dunn’s post hoc test were used for multiple comparisons between independent groups.

## Figures and Tables

**Figure 1 ijms-24-01585-f001:**
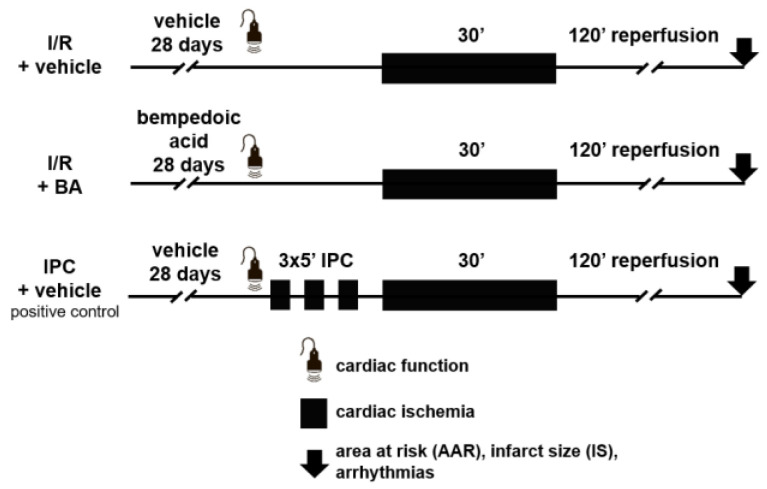
Experimental protocol. Rats were treated with vehicle or bempedoic acid for 28 days per os. After the treatment period, echocardiography was performed. Myocardial ischemia was implemented by occlusion of the left anterior descending coronary artery and area at risk, infarct size and arrhythmias were measured. Ischemic preconditioning (IPC) was used as a positive control. IPC: ischemic preconditioning. I/R: ischemia/reperfusion.

**Figure 2 ijms-24-01585-f002:**
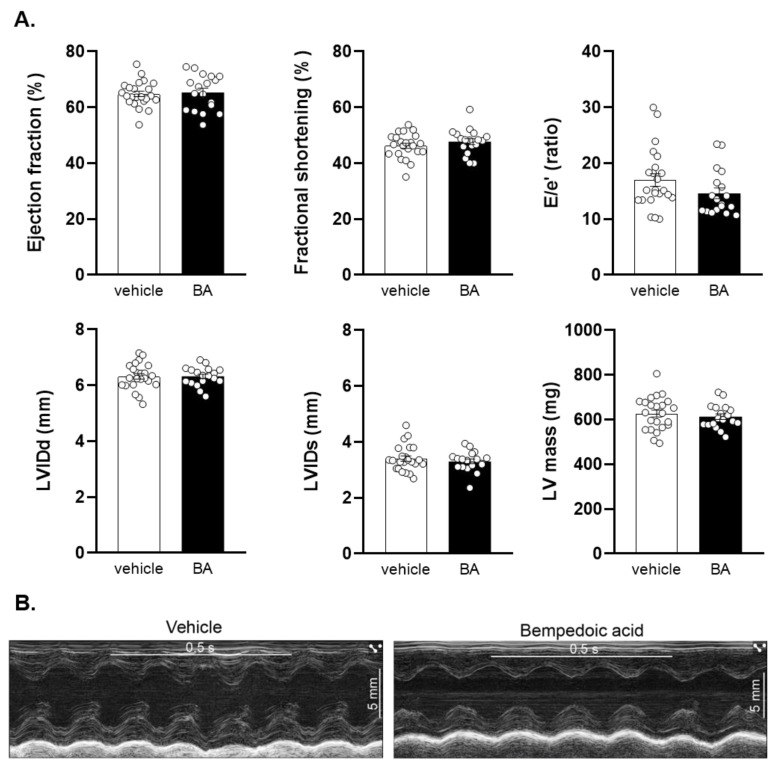
Baseline cardiac function after vehicle or BA treatment. (**A**) Selected parameters of systolic and diastolic function are shown. Chronic treatment with bempedoic acid did not affect functional or morphological parameters measured by echocardiography. Results are presented as means ± SEM. Statistics: Student’s two-tailed *t*-test or Mann–Whitney U-test: n.s., *n* = 18 for BA and *n* = 23 for vehicle treated groups. (**B**) Representative echocardiographic M-mode images of vehicle and BA treated animals. Scale bar: 5 mm. Time stamp: 0.5 s. BA: bempedoic acid. EF: ejection fraction. FS: fractional shortening. E/e’: ratio between peak Doppler blood inflow velocity across the mitral valve during early diastole and peak tissue Doppler of myocardial relaxation velocity at the mitral valve annulus during early diastole. LVIDd: left ventricular internal diameter at diastole. LVIDs: left ventricular internal diameter at systole. LV mass: left ventricular mass.

**Figure 3 ijms-24-01585-f003:**
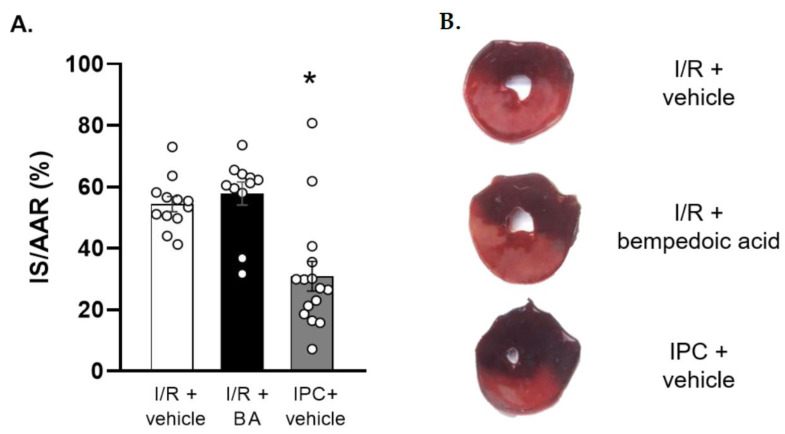
Myocardial infarct sizes. (**A**) Chronic bempedoic acid treatment does not influence infarct size compared to vehicle treated animals, while the positive control IPC significantly reduces it. Results are presented as means ± SEM. * *p* < 0.05 vs. I/R + vehicle and I/R + BA groups, Kruskal–Wallis test, followed by Dunn’s post hoc test, *n* = 11–15/group. (**B**) Representative triphenyltetrazolium-chloride-stained slices. BA: bempedoic acid. IS: infarct size. AAR: area at risk. IPC: ischemic preconditioning. I/R: ischemia/reperfusion.

**Figure 4 ijms-24-01585-f004:**
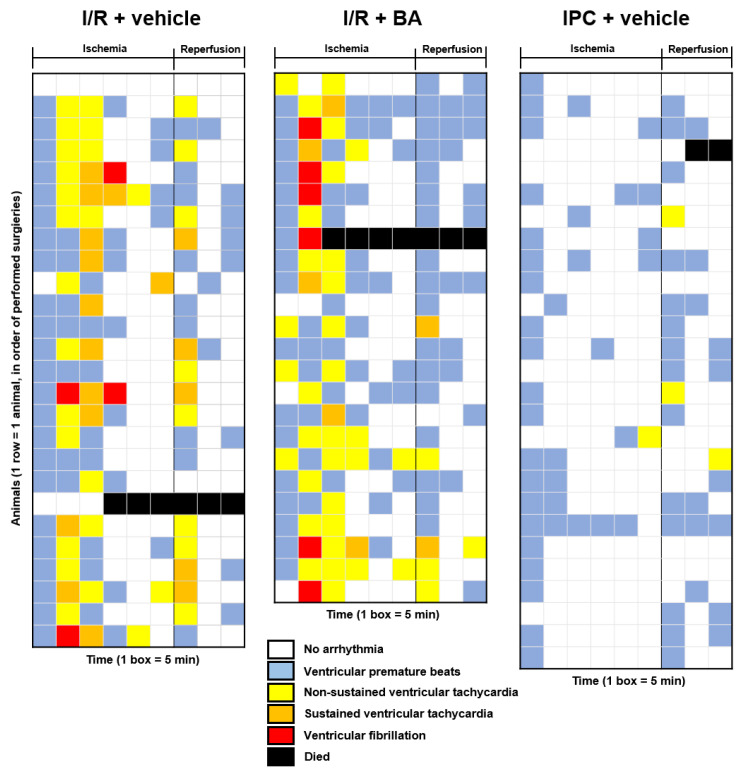
Arrhythmia maps. The most severe arrhythmia is shown in 5 min intervals during the 30 min of ischemia and at the first 15 min of reperfusion, in the order of performed surgeries. Each row represents arrhythmias of one animal. Each box shows 5 min periods colored according to the most severe arrhythmia. Animals that died during IPC (*n* = 1) are not shown on the arrhythmia map. BA: bempedoic acid. I/R: ischemia/reperfusion. IPC: ischemic preconditioning.

**Figure 5 ijms-24-01585-f005:**
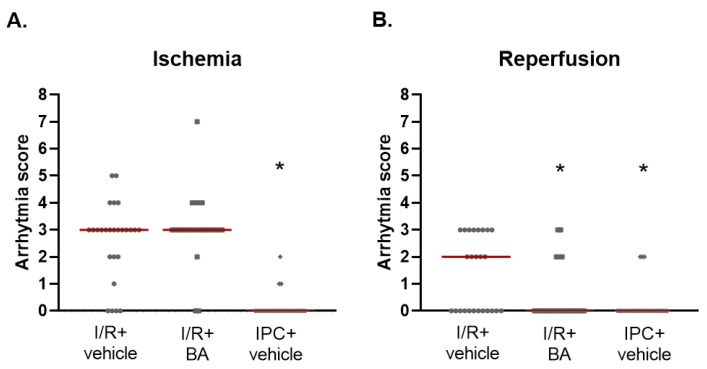
Arrhythmia scores during ischemia (**A**) and the first 15 min of reperfusion (**B**). 28 days of bempedoic acid treatment does not affect arrhythmia score during ischemia but reduces it during reperfusion. Results are presented as median (red line) with individual data points. * *p* < 0.05 vs. I/R + vehicle group, Kruskal–Wallis test, followed by Dunn’s post hoc test, *n* = 23–27/group. BA: bempedoic acid. IPC: ischemic preconditioning. I/R: ischemia/reperfusion.

## Data Availability

Data is contained within the article and Appendix A.

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
