# Peer review of "Effects of Bempedoic Acid in Acute Myocardial Infarction in Rats: No Cardioprotection and No Hidden Cardiotoxicity"

_ijms, 2023, doi:10.3390/ijms24021585_

Round 1
Reviewer 1 Report
Interesting, well-planned research. The reported results are of great clinical value. The order of the individual sections of the manuscript seems a bit strange to me - with methods presented at the end. The only change to consider is changing the sections order and placing 'Methods' before 'Results'.
Author Response
General comment: Interesting, well-planned research. The reported results are of great clinical value. The order of the individual sections of the manuscript seems a bit strange to me - with methods presented at the end. The only change to consider is changing the sections order and placing 'Methods' before 'Results'.
Response:
Thank you for the overall positive feedback on our manuscript. Regarding the order of the individual sections, it is set by the Author Guidelines of the Journal, therefore, after consulting with the handling editor of our manuscript, we are not able to change it.
Reviewer 2 Report
I have reviewed the manuscript entitled 'Effects of bempedoic acid in acute myocardial infarction in rats: 2 no cardioprotection and no hidden cardiotoxicity'.
Several score systems should be formed in order to detect cardiotoxicity in patients using ECG and echocardiography. Thus, this scores can be used to detect the hearts which were exposed to toxicity. Please consider citing 'A simple formula to predict echocardiographic diastolic dysfunction-electrocardiographic diastolic index' . These scores can also be used to test the drugs for their cardiotoxicity effects.
Lipid lowering drugs can be presented as they have several cardiotoxic effects. These drugs should be explained to patients that they have no toxic effect on myocytes. The wrong info on internet has been an issue in a recent paper please also add a short section to the discussion about it. Consider citing 'Evaluation of websites reached using Google in the modern digital era related to approach to cholesterol'.
Author Response
I have reviewed the manuscript entitled 'Effects of bempedoic acid in acute myocardial infarction in rats: no cardioprotection and no hidden cardiotoxicity'.
Comment 1: Several score systems should be formed in order to detect cardiotoxicity in patients using ECG and echocardiography. Thus, this scores can be used to detect the hearts which were exposed to toxicity. Please consider citing 'A simple formula to predict echocardiographic diastolic dysfunction-electrocardiographic diastolic index' . These scores can also be used to test the drugs for their cardiotoxicity effects.
Response 1: Thank you for the suggestion to use this score system. Unfortunately, in our study this scoring is not applicable, as we only recorded one lead of electrocardiogram during the surgeries, while the score system requires measurements from multiple leads. Nevertheless, we added a sentence to the discussion section on the use of ECG-based score systems as a method of testing for drug-induced cardiotoxicity on page 7 lines 205-209 as follows: „Further studies, using preclinical arrhythmia models, such as ex vivo simulated I/R injury [3], animals with reduced repolarization reserve [25], [26], or electrocardiogram-based scoring systems [27] may be required to further demonstrate the cardiac safety of this novel antihyperlipidemic drug.” Furthermore, we added a sentence to the discussion section on the lack of effect of BA on myocardial function on page 7 lines 201-202 as follows: “BA treatment did not affect systolic or diastolic myocardial function, further showing the good safety profile of BA.”
Comment 2. Lipid lowering drugs can be presented as they have several cardiotoxic effects. These drugs should be explained to patients that they have no toxic effect on myocytes. The wrong info on internet has been an issue in a recent paper please also add a short section to the discussion about it. Consider citing 'Evaluation of websites reached using Google in the modern digital era related to approach to cholesterol'.
Response 2: Thank you for your comment. Accordingly, we have amended the „introduction” section with the suggested citations on page 2 lines 48-49 as follows: „Although there is a clear cardiovascular benefit of statin use, non-adherence to statin therapy is quite common [6], in part due to misinformations on lipid-lowering therapy [7]”
Round 2
Reviewer 2 Report
The article is fluent and acceptable in its current form.